# pyBumpHunter : A model independent bump hunting tool in Python for High Energy Physics analyses

Louis Vaslin[1], Samuel Calvet[1], Vincent Barra[2], and Julien Donini[1]

[1]LPC, Université Clermont Auvergne, CNRS/IN2P3, Clermont-Ferrand; France
[2]Université Clermont Auvergne, CNRS, LIMOS, UMR 6158, Clermont-Ferrand; France

April 6, 2023

**Abstract**

The BumpHunter algorithm is widely used in the search for new particles in High Energy Physics analysis. This algorithm offers the advantage of evaluating the local and global p-values of a localized deviation in the observed data without making any hypothesis on the supposed signal. The increasing popularity of the Python programming language motivated the development of a new public implementation of this algorithm in Python, called pyBumpHunter, together with several improvements and additional features. It is the first public implementation of the BumpHunter algorithm to be added to Scikit-HEP. This paper presents in detail the BumpHunter algorithm as well as all the features proposed in this implementation. All these features have been tested in order to demonstrate their behaviour and performance.

## Introduction

The BumpHunter algorithm was originally proposed by [1]. Its aim is to search for a localized deviation in a histogram with respect to a reference, and to provide its statistical significance after having accounted for the Look Elsewhere Effect [2]. One of the advantages of this method is that there is no need to assume any signal model, making the search for deviations model agnostic. Such algorithms can have a lot of applications, and in particular in High Energy Physics (HEP) when searching for new particles. Indeed, one of the most used strategies for such searches is to look for a localized excess in the distribution of a variable of interest. For this reason, several examples of analysis using this algorithm can be found in the literature [3][4].

In recent years, the Python programming language has gained a lot of importance in HEP analysis, especially with the emergence of advanced libraries dedicated to data analysis and Machine Learning. For this reason, we decided to propose the first public implementation of the BumpHunter algorithm in Python to be included in Scikit-HEP [5]. The objective we aim to reach is to propose an accessible tool that enables the use of the BumpHunter algorithm in a Python workflow, and that also provides several new features extending the original algorithm. The source code of this package called pyBumpHunter can be freely accessed [6]. Examples of use-cases are available in the main repository.

In this paper we present the basics of the BumpHunter algorithm as well as all the extensions provided, including a signal injection based sensitivity test, an extension of the algorithm to 2D histograms and a background normalization procedure. The multi-channel combination method proposed in this implementation is similar to the one originally proposed in [1]. All the tests of the different extensions have been realized with toy data generated using Numpy random generator [7]. Results were obtained using pyBumpHunter version 0.4.0. All the codes used to perform the tests presented here can be found on our GitHub repository [8].

# 1 The BumpHunter algorithm

The BumpHunter algorithm searches for a deviation in the binned distribution (histogram) of an observed variable obtained from a counting experiment that follows the Poisson statistic. When searching for a deviation, the observed data distribution is compared to a reference histogram with the same binning. The reference corresponds in general to the expected distribution according to a background model. The model can be either simulated following a known theory or determined experimentally. In order to test the presence of a deviation, the algorithm scans the distributions using a sliding window of variable width. This scan window tests all the interval positions for every allowed width, defined as an integer number of bins. In practice, the minimum and maximum scan widths, as well as the scan step, can be set by the user. For example, when scanning a 40 bins histogram with a scan window width varying between 1 and 5 bins, a total of 190 intervals are being tested. Figure 1 illustrates the scanning procedure.

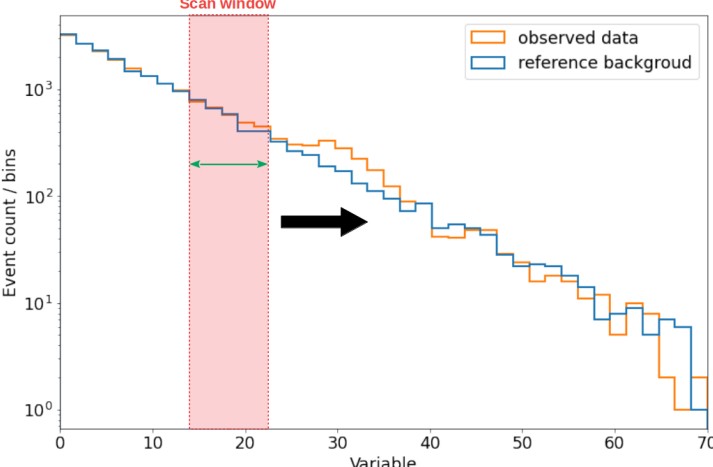

Figure 1: Scanning procedure performed by the BumpHunter algorithm. The red rectangle shows the interval that is currently being analyzed and the black arrow represents the motion of the scan window over the histogram range. In this example, the scan width is 5 bins and the scanned distributions have 40 bins.

In order to assess the presence of a deviation for a given interval, a local p-value is computed. The larger the deviation, the smaller the p-value. The looked-for deviation can be either a localized excess or a localized deficit of observed data with respect to the reference. For an excess, the local p-value is defined as follows:

$$p(D, B) = \begin{cases} \Gamma(D, B) & \text{if } D \geq B \\ 1 & \text{if } D < B \end{cases}, \tag{1}$$

where D and B correspond to the number of observed data and reference background inside the tested interval respectively, and $\Gamma$ is the normalized lower incomplete version of the Gamma function.
For a deficit, the local p-value is defined as follows:

$$p(D, B) = \begin{cases} 1 & \text{if } D > B \\ 1 - \Gamma(D + 1, B) & \text{if } D \leq B \end{cases}. \tag{2}$$

It is possible to choose whether we look for an excess or a deficit. Once the local p-value of all tested intervals has been evaluated, the result can be summarized in the form of a tomography plot as shown on figure 2. This plot shows all the intervals for which an excess of data with respect to the reference background has been observed, together with their associated p-value. The same can be obtained when looking for a deficit.

The local p-value can then be transformed into a BumpHunter test statistic value. The test statistic $(t)$ is defined such that it increases monotonically when the amount of data in excess (or deficit) increases :

$$t = -\ln(p). \tag{3}$$

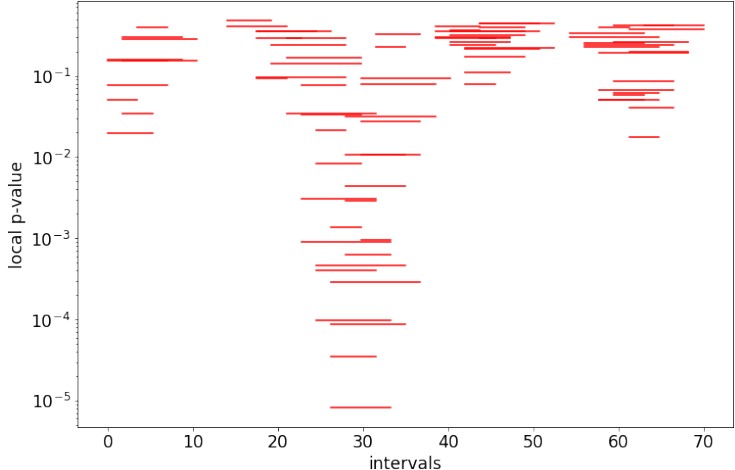

Figure 2: Tomography plot obtained with the BumpHunter algorithm. The position and width of every red line correspond to the tested intervals for which a excess of data with respect to the background was observed, and their $y$ coordinates correspond to the local p-values.

In HEP analysis, the results are often represented in terms of statistical significance ($\sigma$). This significance corresponds to how much the data deviates from the central value of a normal distribution as a number of standard deviations. The significance can be computed from the p-value using the inverse cumulative density function, also called percent point function (ppf), of the normal distribution:

$$\sigma = \mathrm{ppf}(1 - p(D, B)). \tag{4}$$

When computing the local significance, $p(D, B)$ is the local p-value as defined in equation 1 or 2. By convention, in case a deficit is observed, the sign of the significance will be flipped.

Once all the intervals have been tested, the one corresponding to the most significant deviation of the observed data with respect to the reference is defined as the interval with the smallest local p-value. However, the higher the number of bins, the more likely it is to observe a local deviation, so a small local p-value, due to statistical fluctuations. This is the so-called Look Elsewhere Effect [2]. In order to account for this effect, one can compute a global p-value that relates the observed local deviation with what can be expected given the statistical fluctuations in the reference background[1].

To obtain a global p-value, BumpHunter uses thousands of pseudo-data distributions generated from the reference histogram. In the proposed implementation, the pseudo-data histograms are generated by smearing each bin count of the reference histogram according to a Poisson law. The scan procedure is then applied to every pseudo-data histogram as if it was the observed data histogram, resulting in a minimum p-value being associated to each histogram. With this procedure, the algorithm produces a minimum local p-value distribution that corresponds to a reference background only hypothesis. Finally, the global p-value can be computed by comparing the local p-value obtained from the observed data with the background only distribution. In other words, the global p-value corresponds to the p-value of the local p-values and is computed as in equation 5.

$$\mathrm{p} = \frac{N_{>data}}{N_{tot}} \tag{5}$$

With $N_{>data}$ the number of pseudo-data distributions that has a local p-value smaller than the one of data (i.e a larger test statistic value), and $N_{tot}$ the total number of generated pseudo-data distributions. One should note that, with this definition, the global p-value can be higher than 0.5, producing a negative global significance.

---

[1]The method proposed by the BumpHunter algorithm relies purely on frequentist considerations, although there also exists different methods relying on a Bayesian approach [9]

In this case, this negative value should not be interpreted as a deficit in the data, but as a deviation which is less significant than the median deviation observed in the background-only pseudo-data distributions. However, when considering a search for a deficit, the sign of the significance is flipped, so it will be negative in case the global p-value is smaller than 0.5 and positive otherwise. Figure 3 shows an example of BumpHunter test statistic distribution obtained by running the algorithm on toy data and reference distributions. The distribution of test statistic value related to the pseudo-data was obtained by generating and scanning 80000 pseudo-data distributions.

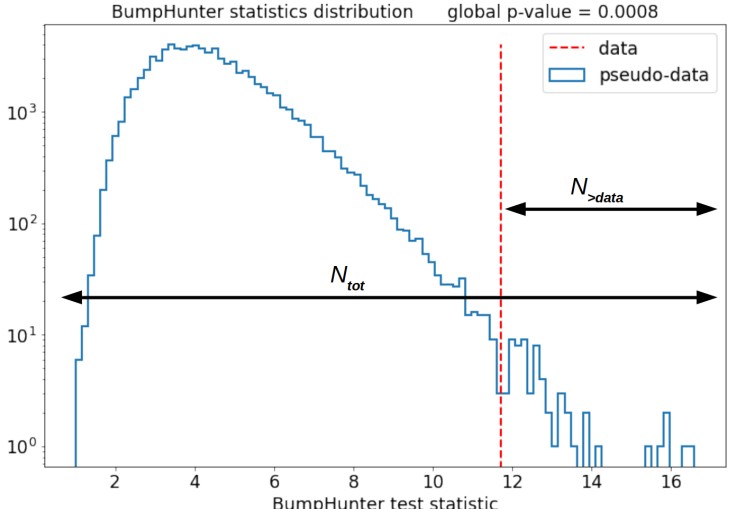

Figure 3: Distribution of BumpHunter test statistic value obtained for the pseudo-data (blue). The dashed red line correspond to the value of test statistic associated to observed data. The arrows illustrate how the global p-value is computed in equation 5. The global p-value is 0.0008, corresponding to $3.16\sigma$.

This way of computing the global p-value is efficient to account for the look else-where effect, but has one limitation: the minimum non-null global p-value that can be computed depends on the number of pseudo-data distributions that were generated. When the count of pseudo-data distributions for which the test statistic passes the threshold ($N_{>data}$) becomes small, the obtained global p-value becomes inaccurate. In the extreme case when $N_{>data}$ becomes exactly 0, it causes the global significance to become infinite. A solution could be to increase the number of generated pseudo-data distributions. However it takes more than 30 millions pseudo-data distributions to obtain a global significance of $5\sigma$ with enough accuracy. Such an amount requires a computation time of several hours, even when parallelizing on many CPU cores. Another approach to address this problem is discussed in [10].

This is the original algorithm proposed in [1]. All the other features presented in the following sections are extensions of this algorithm we propose.

## 2 Signal injection and sensibility test

### 2.1 Principle

The objective here is to assess the sensitivity to a given signal hypothesis that deviates from expectation. In order to perform this test, we need a signal histogram that models the hypothesis to be tested. Since BumpHunter aims at finding localized deviation, the signal must add either a localized excess or deficit with respect to the reference histogram. In addition to the signal histogram, we also need to give an estimation of the amount of signal that is expected according to the hypothesis.

The sensitivity test algorithm is the following. A data histogram is generated using the reference and the injected signal histograms. Depending on the importance of the injected signal, we define the signal strength as the ratio between the number of

injected signal events ($N_{injected}$) and the number of expected signal events ($N_{expected}$):

$$\mu = \frac{N_{injected}}{N_{expected}}. \tag{6}$$

The BumpHunter algorithm can then be applied using the histogram obtained after injection in place of the observed data. In order to evaluate the uncertainty on the global significance, the scan of the data is repeated many times by varying the histogram bin contents with a Poisson law. It will allow the algorithm to produce a local p-value distribution corresponding to a background+signal hypothesis ($H_1$). Thus, the global significance is defined according to the median of the $H_1$ hypothesis. The lower and upper errors on this value are defined according to the first and third quartiles of the $H_1$ hypothesis.

The procedure is repeated with an increased signal strength at each iteration, such that the global significance starts increasing together with the signal strength. Once the global significance becomes higher than a user defined threshold, the injection stops with a signal strength corresponding to the required sensitivity. In HEP analysis, the rule is that a deviation from expectation must reach a global significance of at least $5\sigma$ in order to claim a discovery, and at least $3\sigma$ for an evidence.

## 2.2 Test

The objective is to illustrate the behaviour of the signal injection procedure and its application to sensitivity tests. The reference background consists of a 1D exponential distribution. The probability density function of the exponential distribution is defined as:

$$\exp(x) = \frac{1}{\lambda} e^{-\frac{x}{\lambda}}, \tag{7}$$

with $\lambda$ the scale parameter, or slope of the distribution. The exponential used for this test has a slope of 8 with $5 \times 10^5$ generated events in the interval [0,70]. The signal is defined as a Gaussian distribution with mean of 20 and standard deviation of 3 with an expected number of signal events set to 1000 (corresponding to $\mu = 1$). The histograms have 50 bins of equal width. The injection is performed with a signal strength that starts at 0.1 and increases with a step of 0.1 at each iteration. The procedure is set to stop when the global significance becomes greater or equal to $3\sigma$. The number of background-only pseudo-data distributions used to compute the global p-value and significance is set to $8 \times 10^4$. The number of background+signal pseudo-data distributions used to evaluate the error bars is set to 200. The result of the sensitivity test is shown on figure 4.

It shows that the global significance increases steadily together with the signal strength until a signal strength of 0.7 is reached (corresponding to 700 injected signal events). For the first three points, the lower error bars go below 0. This is expected for a low value of signal strength where the deviation induced by the presence of the signal is negligible compared to the statistical fluctuations. When reaching a signal strength of 0.7, the median global significance is $3.2\sigma$, which is greater than the required sensitivity.

One should notice that, due to the limitation of the BumpHunter algorithm when the number of pseudo-experiment generated to compute the global p-value is not high enough, the required sensitivity might not be reachable. In practice, this happens when the numerator goes to 0 in equation 5. Thus, the global significance would go to infinity, raising errors in the execution of the code. For this reason, the given global significance is set to be the highest non-infinite significance that can be reached given the number of pseudo-data distributions that were generated. In this case, it should be interpreted as a lower limit and will be represented on the plot with an arrow on the error bar.

# 3 BumpHunting in 2D

## 3.1 Principle

The pyBumpHunter package gives the possibility to look for localized deviations in 2D distributions. Here, a 2D distribution corresponds to the distributions of 2 different

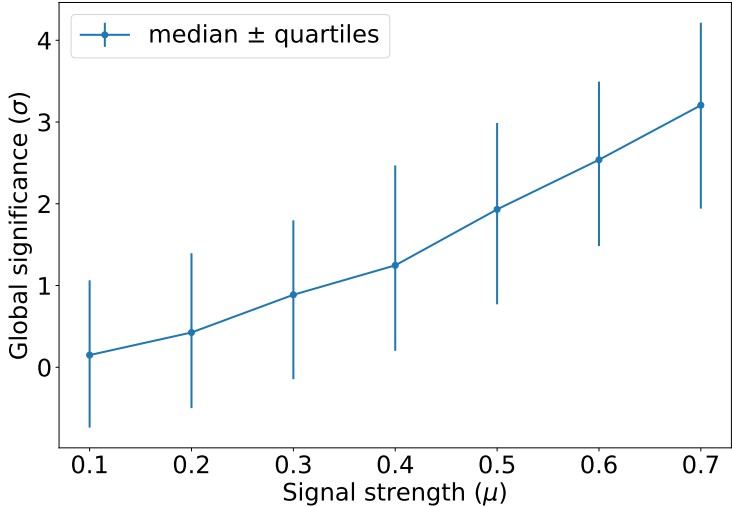

Figure 4: Evolution of the global significance found by BumpHunter as a function of the signal strength. The points correspond to the median global significance and the down and up error bars to the first and third quartile respectively. This is a non-standard definition of uncertainties that will be modified in future release to cover a 68% confidence interval corresponding to $1\sigma$.

observables that may be correlated. A 2D histogram is built by counting the events falling in regions of the 2D plane delimited by 4 bin edges. This extension of the original BumpHunter algorithm applies the same procedure described previously using the 2D histograms. In this case, the interval width is defined according to the two axes of the histogram, resulting in rectangular windows. For each width of the scan window, all the positions in the plane are tested, similarly to what is done with 1D histograms.

## 3.2 Test

The objective of this test is to evaluate the behaviour of the extension of BumpHunter to 2D histograms. The reference histogram consists of a 2D exponential to which a flat random noise was added in order to avoid too many empty bins in the tail. The exponential has a slope of 4 along both axes without correlations. In order to have a smooth histogram for the reference background, a total of 10 millions events have been sampled from the exponential to which $1.5 \times 10^5$ events sampled from a uniform distribution have been added. Then, the resulting 2D histogram was scaled down by a factor 100. The observed data has been built using the same background density, with $10^5$ events sampled from the exponential and 1500 from the uniform distribution. On top of that, a localized excess has been added to serve as a simulated signal. This signal is defined by a 2D Gaussian distribution with mean $\mu = \binom{5}{5}$ and covariance matrix $\sigma = \begin{pmatrix} 1.5 & 0 \\ 0 & 1.5 \end{pmatrix}$. An example of 2D histogram together with its 1D projections are shown on figure 5.

The 2D histograms have 20 bins along both axes, ranging in [0,25], which gives 400 square bins. The width of the scan window has been set to vary between 2 and 5 bins along both axes. The global p-value and significance were computed using $10^4$ pseudo-data distributions. For each tested signal, the scans have been repeated 100 times using data generated with a different seed. This gives an estimate of the variability of the results given by the algorithm. Results are shown on figure 6.

In the top left plot, the dashed lines correspond to the true value of the signal mean. When there is no signal in the data, the position found by the BumpHunter algorithm is on average close to the middle of the histogram range for both x and y with a large variability. This is expected since only random fluctuations can be detected in this case. However, when the number of signal events increases, the average position of the deviation converges toward the true value. This is expected since the algorithm is more likely to recover the true signal position as it becomes more important. When

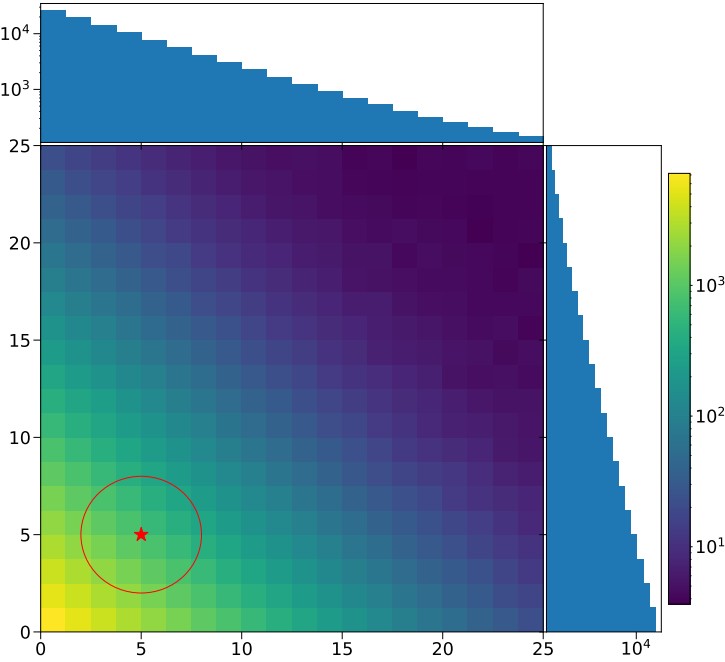

Figure 5: Example of 2D histogram used with BumpHunter. The top and right panels show the projections along the y and x axes respectively. The red star and circle show the injected signal mean and $2\sigma$ contour respectively.

reaching 600 signal events, the position recovered by the algorithm stabilizes slightly over the true position for both x and y. This bias is due to the asymmetry of the background distribution. As there are more background events for small x and y, it is more difficult to recover the tail of the signal in this region, shifting the recovered position toward regions with less background events. When the background is defined as a flat distribution, we verify that this bias disappears.

In the top right plot, the dashed line corresponds to the expected trend if the evaluated number of signal events is equal to the true one. We observe that the evaluated number increases together with the true number. However, we notice that the trend it follows is different from what we would expect as the evaluated number of signal events is underestimated compared to the true value. One explanation is that the algorithm does not recover the entirety of the signal. In particular, the tail of the signal can be covered by the statistical fluctuation in the background. Note also that, in case the width of the scan window is smaller than the real width of the signal, a bias will also appear. It has been observed using a larger signal with the same window width that the bias on the number of evaluated signal events increases.

In the bottom left plot, the test statistic increases as the signal becomes more important as one would expect. The slope between 0 and 500 signal events is smaller than the slope afterward because we are in a regime where the deviation induced by the presence of signal is not very significant compared to the statistical fluctuations in the background.

As expected, the first point in the bottom right plot corresponding to a data distribution with no signal (background only) has a median global significance close to 0 compatible with statistical fluctuations. Then the global significance increases with the number of signal events up to 700 signal events. After this point, the global significance stabilizes around $4\sigma$. This saturation of the global significance means that we reach the limit that can be achieved given the number of available pseudo-data distributions. Spreads also seem to become smaller after 600 signal events, and are null from 800 signal events and after. They are computed by repeating the scan 100 times and taking the first and third quartiles of the global significance. If the error bars go to 0, it means that at least 3/4 of the 100 global significances have reached the saturation value.

In order to evaluate the results of figure 6, we compare them with those obtained with 1D scans as a reference. For this purpose, 1D scans have been performed after

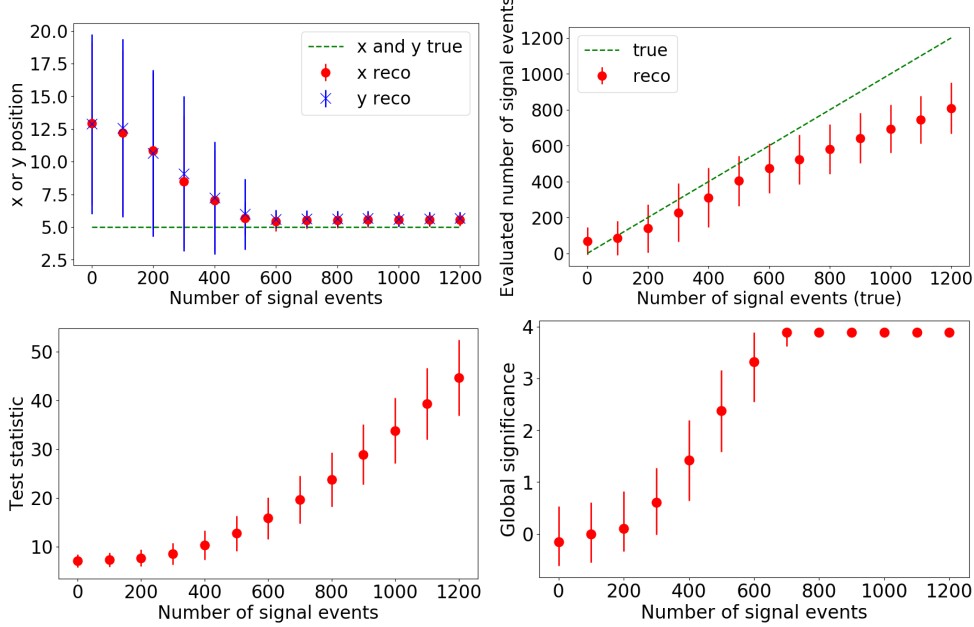

Figure 6: Evolution of the deviation excess (top left), evaluated number of signal events (top right), test statistic value (bottom left) and global significance (bottom right) as a function of the true injected number of signal events for the 2D scans.

projecting the 2D histograms along the x and y axis respectively. The results are presented on figure 7. In this case, the red circles and blue crosses correspond to the scans performed along the x and y axes respectively.

The trend that can be observed in these 4 plots are similar to what was observed for the 2D scans. However, we can notice some differences. First, we observe a small difference in the average position along the x and y axis of the deviation found by BumpHunter. This difference was not visible in the results of the 2D scans, and is not expected since there are no differences between the x and y true positions of the signal. One thing that could explain this difference is differences in the projections of the 2D data histogram due to statistical fluctuations. It could induce a different bias in the position of the signal recovered along the two axes. However, this difference disappears as the recovered position converges toward the true position of the signal. When the deviation induced by the presence of a signal becomes more significant than statistical fluctuations, there is no bias anymore.

Secondly, the bias on the evaluated number of signal events is smaller in the 1D scans compared to the results obtained for the 2D scans. Indeed, for 1400 signal events, the difference between the evaluated average and the true value was around 400 events for the 2D scans, while it is around 200 events for the 1D scans. This difference can be explained by the fact that there is more signal that is not recovered by the algorithm when the signal is spread along multiple axes. By projecting the distributions on one axis, the signal is indeed integrated along the direction of projection. Thus, the bins that had low signal content in the 2D histogram have higher signal content after projection. The bias induced by the loss of signal events in the tail of the deviation is reduced. This shows that the evaluation of the number of signal events is less precise when performing a 2D scan compared to a 1D scan.

Finally, if the trend of the evolution of the test statistics value as a function of the number of signal events is the same, the reached values are not the same. For instance, the average test statistic value obtained for 1400 signal events was around 60 for the 2D scans, while it is below 30 for the 1D scans. This fact goes with the observation that the global significance reaches the saturation limit for 1000 injected signal events for the 1D scans, while it was reached at 800 injected signal events for the 2D scans. This shows that the global significance of the deviation induced by the presence of signal increases faster with the number of signal events when performing a 2D scan compared to a 1D scan. In other words, in the presence of a signal, it is easier to find a significant deviation in the data with respect to the reference background

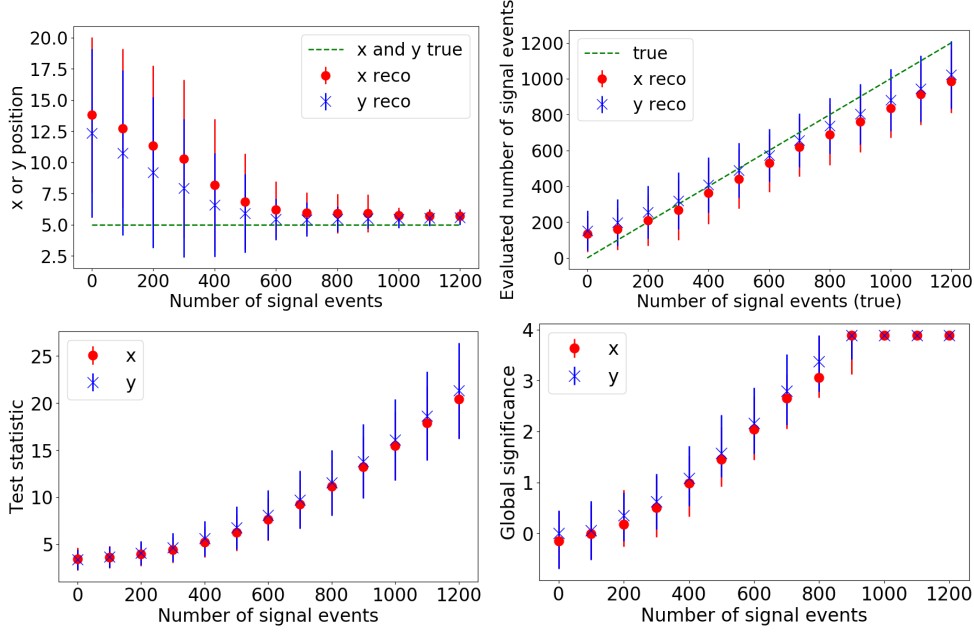

Figure 7: Evolution of the deviation position (top left), evaluated number of signal events (top right), test statistic value (bottom left) and global significance (bottom right) as a function of the true number of signal events for the 1D scans.

with a 2D scan than in a 1D scan. Indeed, if the signal content is integrated along the direction of projection when making a 1D histogram, it is also the case for the background. In the case of a localized signal, the integrated amount of background events in a given bin increases faster than the integrated amount of signal events. The signal has events only in a limited range of the projection axis since it is localized, while the background is spread over the entire range of the projection axis. Thus, the ratio of the number of signal and background events is less favorable when performing a 1D scan, resulting in a smaller local and global significance. We can conclude that, despite the higher bias of the evaluated number of signal events in the deviation, performing a 2D scan when possible is more likely to give a better significance in the presence of a localized signal.

## 4 Side-band normalization

### 4.1 Principle

In High Energy Physics, it is common to use simulations in order to estimate the reference background. In some cases, the exact normalization of the background with respect to available data is not well known, inducing a difference in scale between the data and reference background histograms. In order to handle this situation, py-BumpHunter provides a solution to normalize the reference histogram to the observed data while avoiding any bias that might be induced by the presence of a deviation.

The procedure is the following. For each tested position and width of the scan window, the number of reference background events is corrected using the ratio of observed and expected numbers of events outside the interval. Thus, the number of reference background events in the interval becomes:

$$B' = \frac{D_{tot} - D}{B_{tot} - B} \times B, \tag{8}$$

where $D_{tot}$ and $B_{tot}$ are the total numbers of events in the observed and reference histograms respectively, and D and B the numbers of observed and reference events in the tested interval respectively. This correction of the reference is computed for each interval during the scan. One should note that, in case of discrepancy between the reference and observed histograms outside of the tested interval, the rescaling will be affected. A simple example is the case where the width of the tested interval doesn't

fully cover the deviation. In this case, part of the excess (or deficit) of observed data will be included when computing the normalization ratio in equation 8. This will have the effect of lowering the local significance associated to this interval, favoring intervals that fully cover the deviation range.

Depending on the shape of the distributions that are being scanned, it has been found that this side-band normalization procedure can penalize the global p-value. This bias is induced by the dynamic nature of the normalization procedure and occurs in cases when there is no significant deviation due to the presence of a signal, as it is the case for the background-only pseudo-data histograms. During the scan, the algorithm tends to select the interval for which the normalization factor is the most favorable. In the search for an excess, it corresponds to the lowest normalization factor that lowers the number of reference background events. This bias on the evaluation of the reference background is translated to the test statistic distribution of the background-only hypothesis. In order to reduce this bias, an additional parameter is introduced, allowing to define an excluded region on each side of the scanned histograms. This region is excluded for the scan itself, so the algorithm will not look for deviations in this region. However, it is kept when computing the normalization scale for each tested interval. With pyBumpHunter, the width of the excluded region can be set by the user. The figure 8 illustrates the principle of the side-band normalization procedure. Side-bands are used in the original proposal of the algorithm but do not treat the problem of the floating normalization of the reference background, as we propose.

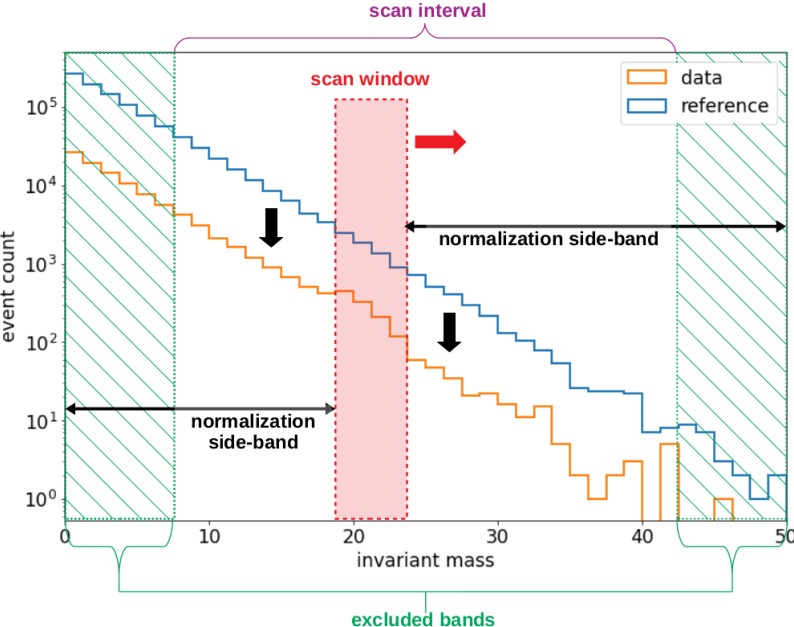

Figure 8: Figure illustrating the principle of the side-band normalization method applied to two histograms. The normalization side-bands are represented by the horizontal two-sided black arrows. The excluded bands are represented by the green dashed zones on each side of the spectrum. The scan interval in which the bump hunt is performed corresponds to the histogram's full range minus the excluded bands.

## 4.2 Test

The objective is to evaluate the effect of background normalization, with and without specific excluded regions for the scan range. The background is defined as a 1D falling exponential distribution of slope 4. A total of $10^6$ events have been generated for the reference background distribution, against only $10^5$ for the data distribution. The signal is defined as a Gaussian distribution of mean $\mu = 6$ and standard deviation $\sigma = 1$. The number of injected signal events varies from 0 to 1800 with a step of 200. The histograms have 40 bins of equal width in the range [0,20]. The scan window width has been set to vary between 2 and 6 bins. The global p-value and significance

have been obtained using $6 \times 10^4$ pseudo-data distributions. In order to evaluate the effect of using side-band normalization, three cases have been considered: scans without using side-band normalization (use true normalization factor instead), scans using side-band normalization with a excluded region width of 0 bins, and scans using side-band normalization with a excluded region of 5 bins (at the beginning and end of the histogram range). Results are shown on figures 9.

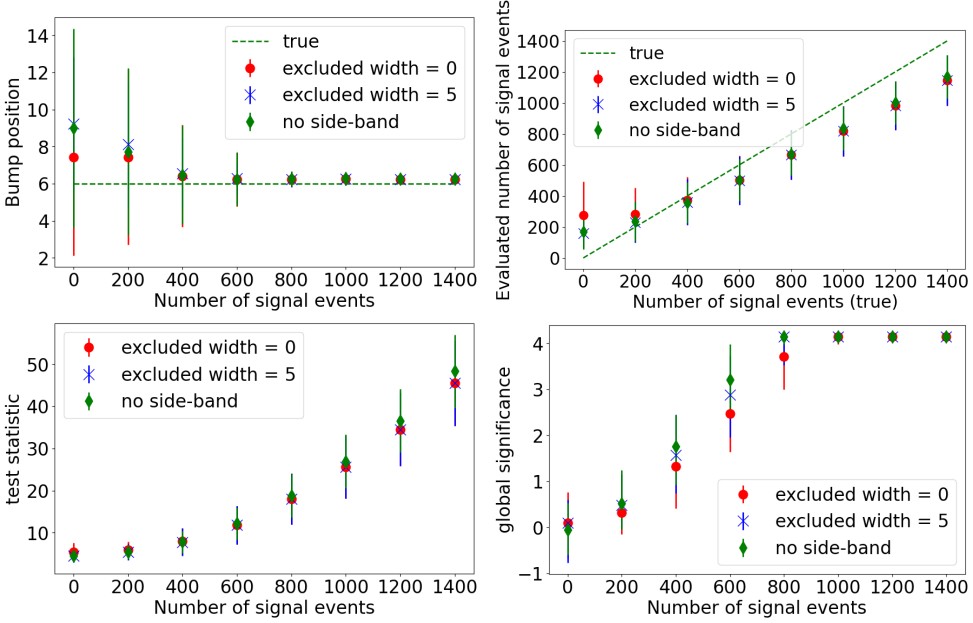

Figure 9: Evolution of the deviation position (top left), evaluated number of signal events (top right), test statistic value (bottom left) and global significance (bottom right) as a function of the true number of signal events. The red circles and blue crosses correspond to the side-band normalization tests with an excluded region width of 0 and 5 bins respectively, the green diamond correspond to the tests without side-band normalization.

On the top left plot, the evolution of the average deviation position shows a behaviour similar to what was observed in section 3.2. However, for the test using side-band normalization without an excluded region, the average position of the deviation is shifted towards lower values when there is no signal in the data. This is related to the bias presented in the section 4.1. In the region with more statistics, the bias in the computation of the local p-value is more important. Thus, the algorithm tends to select intervals in higher statistics regions when there is no signal in the data. Using the excluded region reduces this bias by including a constant normalization area in which the algorithm doesn't look for deviations.

On the top right plot, the evolution of the evaluated number of signal events follows the trend observed previously. We also see that there are no differences induced by the use of side-band normalization, except for the very first point where the average is not the same when there is no excluded region.

The evolution of the test statistic value in the bottom left plot shows the expected behaviour. However, starting from 1000 signal events, we can observe that the average values obtained with side-band normalization deviate from the one obtained without. The difference observed for 1400 signal events is rather small (less than 10%), but it could have an impact on the local and global significance. Also, the average value obtained using side-band normalization isn't affected by the width of the excluded region, as long as they don't overlap with the signal area.

The evolution of the global significance in the bottom right follows the same trend as we have seen before. However, when the side-band normalization is used, the global significance doesn't increase as fast as when the normalization is known and fixed beforehand. Thus, for 600 signal events, the bias reaches almost $1\sigma$ when there is no excluded region. However, this bias is reduced to around 10% of difference with the result obtained without side-band normalization when using an excluded region of

5 bins. This is related to the observation made in the evolution of the mean deviation position. Including a constant normalization region which is not dynamically set reduces the bias on the local p-value associated to pseudo-data histograms. Thus, the global p-value is less penalized than in the case where no excluded regions are used.

# 5 Multiple channels combination

## 5.1 Principle

In High Energy Physics, the same physical process, such as the production and decay of a particle, can be studied in several different signatures. Thus, for each possible final state and applied selections, different channels can be defined. In order to extract the most information from the available observed data, a possible way is to combine all the channels. pyBumpHunter gives the possibility to combine multiple channels in order to obtain a combined local and global significance. In this case, the algorithm takes multiple data and reference histograms, one pair of each per channel. The procedure to look for the most significant localized deviation in a given channel is the same as described previously. In order to combine the results of each individual channel, pyBumpHunter uses the following method.

The interval corresponding to the most significant deviation is defined as the intersection of the intervals selected in each individual channel. In case there is no overlap between all the individual intervals, the algorithm considers that there is no localized deviation that is consistent across all channels. Thus, the combined local p-value is set to one, which corresponds to no deviations observed. In case there exists a non-empty intersection between all selected intervals, the combined local p-value is defined as the product of the local p-values associated with all the selected intervals. This method is applied to both the observed data and the pseudo-data generated from the reference histograms. Then, the global p-value and associated significance are computed from the distribution of combined local p-values.

This combination method can be applied to histograms that have a different binning for every channel. It also requires that the deviation is located at the same position in every channel, so they must have compatible ranges. It also implies that the same variable should be used in all channels, and that non consistent deviations are automatically rejected. This is the first proposed solution, different combination algorithms are currently under study.

## 5.2 Test

The objective is to evaluate the behaviour of the combination method defined in section 5.1, and to compare it with a single channel scan. The reference background is defined as two 1D exponential distributions with a slope of 4.5, one per channel. The data is made from the same exponential as the reference to which Gaussian signals were added with mean $\mu = 8.0$ (both channels) and width $\sigma = 1.25$ in the first channel and $\sigma = 1.5$ in the second one. In the first channel, $1.25 \times 10^5$ background events are generated, against $1.5 \times 10^5$ in the second channel. The number of injected signal events varies from 0 to 1200 with a step of 150 in both channels. Thus, signal over background ratios are different in each channel. However all histograms have the same 50 bins in the range [0,40] in order to be able to compare the result with a simple scan on the sum of the two channel histograms. The width of the scan window has been set to vary between 1 and 5 bins. The global p-value and significance have been computed using $6 \times 10^4$ pseudo-data distributions. As previously, the test is repeated 100 times per point to evaluate the spread. The results are shown on figure 10.

The evolution of the average deviation position (top left) behaves as in the previous tests. However, the average position seems to converge toward the true value faster when performing a single-channel scan. Indeed, in case no consistent deviation is found across channels, the position of the deviation falls back to the middle of the histogram range (20 in this case) with a significance of 0. This implies a shift of the average position toward higher values.

The average evaluated number of signal events (top right) stays at 0 for the two first points in multi-channel. When there is no significant deviation induced by the

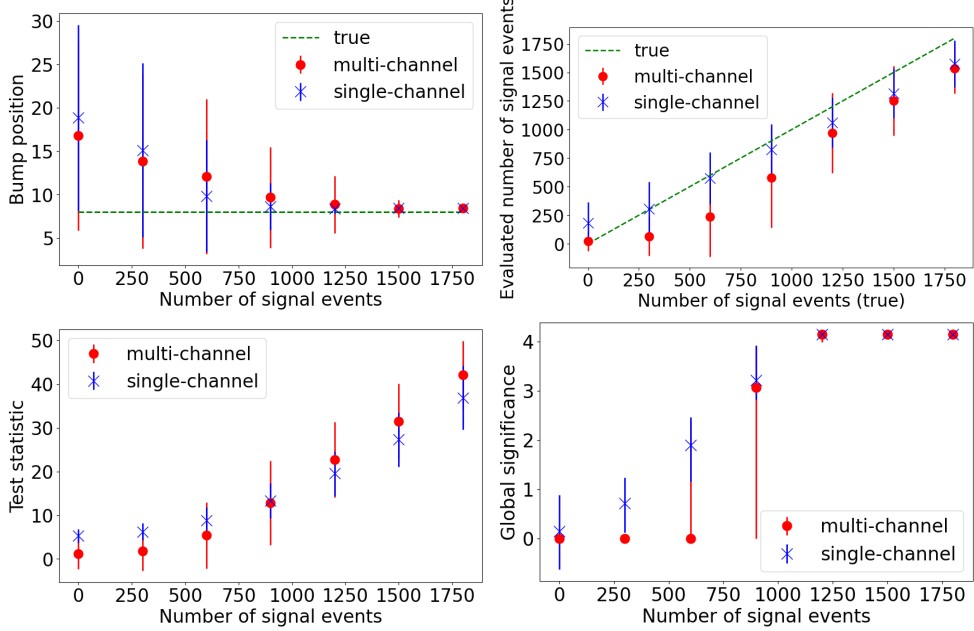

Figure 10: Evolution of the deviation position (top left), evaluated number of signal events (top right), test statistic value (bottom left) and global significance (bottom right) as a function of the true number of signal events for the multi-channel and single-channel tests (red and blue respectively).

presence of a signal, most of the deviations in individual channels don't overlap with each other. Thus, the algorithm considers that there is no consistent deviation and concludes that there is no signal. However, as the number of signal events increases, the multi-channel trend catches-up with the single-channel one, and becomes identical after reaching 750 signal events per channel (1500 in total).

The evolution of the test statistic value (bottom left) shows the expected behaviour. As for the evaluated number of signal events, the average test statistics is close to 0 in multi-channels for the first two points. The explanation is the same as previously. However, after 450 signal events per channel (900 in total), the average test statistic value becomes higher in the multi-channel case. This shows that, when there is a consistent deviation due to the presence of a signal, the test statistics (and so the local significance) is enhanced by the combination of the two channels.

Finally, the evolution of the global significance (bottom right) is similar to what was observed in section 4.2. When there is no signal, the average global significance is 0, as expected. However, as the number of signal events increases, the median global significance increases progressively in the case of a single channel scan. In multi-channel, the median significance stays at 0 until there are 450 signal events per channel (900 in total). This difference is due to the asymmetry of the global significance distribution in multi-channel. When there is no consistent bump across all channels, the global significance is set to 0. It creates a spike at 0 that biases the median downward, similarly to what was observed for the average bump position. The same goes for the quartile for the global significance distribution, hence the large error bars. Thus, in order to have a non null global significance, every channel must be sensitive enough to the presence of signal. Otherwise, the algorithm will most probably claim that it didn't find any deviation. Thus, if the significance can be enhanced in the case a consistent bump is found, the mandatory overlap condition can also penalize it. As mentioned at this end of section 5.1, alternative combination methods are under study in order to solve this issue. Notably the possibility to combine multiple global p-values using techniques based on [11].

# Conclusion

BumpHunter is a useful algorithm that can have several applications in HEP. By evaluating both the local and global p-value of a localized deviation in a binned distribution, it allows for model independent searches for new physics signatures. The pyBumpHunter package was developed for anyone using an analysis workflow based on Python language. This tool is public, easy to handle and has been accepted in Scikit-HEP. In addition to the basic functionalities presented in [1], pyBumpHunter also provides several extensions of the algorithm, including signal injection based sensitivity check, side-band normalization, 2D scans and multi-channel combination. All of these features have been tested in order to evaluate their behaviour and performance.

The results show the capacity of the tool to find the potential presence of a signal with good accuracy and efficiency. The test of the 2D scans also shows the power of this extension of the original algorithm in terms of reached significance, both local and global. However, the tests also show some limitations inherent to the method, such as the saturation of the global significance due to the limited number of pseudo-data distributions. We have also shown that the multi-channel combination algorithm and the side-band normalization procedure can be useful in some use-cases, despite some limitations.

Meanwhile, pyBumpHunter is still in active development. Many of the problems mentioned previously will be addressed in future releases. Also, new features, such as the processing of systematic uncertainties, might be added.

# Acknowledgement

Louis Vaslin acknowledges the support received by the French government IDEX-ISITE initiative 16-IDEX-0001 (CAP 20-25).

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
