# Peer review of "pyBumpHunter: A model independent bump hunting tool in Python for High Energy Physics analyses"

_SciPost Physics Codebases, doi:SciPost Phys. Codebases 15 (2023) , SciPost Phys. Codebases 15-r0.4 (2023)_

## Round 2 · Referee Report · Mohamed Rameez (Referee 1) · 2023-1-2

Strengths

  1. Comes with working code and discusses working examples.
  2. Python implementation of an existing tool.

Weaknesses

  1. Quite narrow in focus.
  2. Appears to reinvent the wheel at times and not sufficiently general in its treatment. This is very much a paper along with a very useful code, rather than a general discussion of the statistical issues.
  3. Doesnt cite a sufficient number of related papers and concepts in statistics literature.

Report

I'd recommend it for publication after some significant revision.

Requested changes

This is a nice paper. I would like to thank the authors for this tool, which is a valuable service to the community. The unit tests on github increase confidence in the immediate use of this tool as a pythonic replacement for BumpHunter. My comments are minor, and merely suggestions:

1. As the tool is aimed at the HEP community which primarily relies on frequentist statistics, the level of detail is fine. However, a more generalized treatment based on Bayesian fundamentals would definitely increase the conceptual and contextual clarity of the paper. In particular, many of the trends in going from univariate to multivariate (bias on recovered signal events becoming worse) have straightforward analogues that have already been reported in literature. For eg, see the difference between the univariate and multivariate KL loss terms discussed in https://www.ncbi.nlm.nih.gov/pmc/articles/PMC6118314/ after equation 4b. I wouldn’t insist on this, since it could be a lot of work and would only improve the paper in some formal sense, but right now to some statisticians it may appear that much of what is being discussed at length in the paper is already known.

2. It would be nice to generalize the discussion immediately after eq 5 to also the case of global significances of deficits. i.e. if a local significance is negative because it’s a deficit, in a situation where the deficit is less significant than the median deficit observed in background-only pseudo-data distributions, will the global significance now be positive? If yes, this seems like a choice of convention, and might benefit from a unified sign convention treatment for global significances for both excesses and deficits.

3. Is the statement in paragraph 2 of page 4 (“When the value of the test statistic of the data falls very far in the tail of the background-only distribution, available statistics to compute N>data becomes very low and the obtained global p-value is not significant. In the extreme case when the value of the test statistic of the data is out of the range of the pseudo-data histogram, it is not possible to compute a global p-value and sig- nificance anymore. ) strictly correct? This may just be semantics, but to me it looks like if there is not sufficient statistics to compute N>data, it will be zero and since the denominator is nonzero, you will get a p value of exactly zero, which will appear to be infinitely significant. This is artificial of course. Perhaps we both mean the same thing, but please consider rephrasing.

4. Section 2.1, page 5, paragraph 1 – “The lower and upper errors on this value are defined according to the first and third quartiles of the H1 hypothesis.” – It’s not clear to me why quartiles were chosen here, shouldn’t values corresponding to the central 68% interval be chosen? This may just be convention. If that is the case I’d request the authors for further clarification in the text.

5. Is the side band normalization procedure carried over from BumpHunter? Or something introduced newly here? If it’s the former I would request that this be made explicit. If It’s the latter I think a longer and more technical discussion is necessary.

  • validity: good
  • significance: ok
  • originality: low
  • clarity: good
  • formatting: good
  • grammar: excellent

Author:  Louis Vaslin  on 2023-02-15  [id 3359]

(in reply to Report 1 by Mohamed Rameez on 2023-01-02)
Category:
answer to question
correction

Dear Mohamed Rameez,

Many thanks for reading our paper in detail and for all your comments.
They have been very useful to improve the quality of our paper and we took most of them into account.
Please see below our answers to your questions and comments.
We uploaded a revised version of the paper on arXiv featuring all the mentioned modification (https://arxiv.org/pdf/2208.14760v3.pdf).

1.
We have found a paper that proposes a Bayesian view of the Look Elsewhere Effect.
https://iopscience.iop.org/article/10.1088/1475-7516/2020/10/009/meta https://arxiv.org/abs/2007.13821
This reference has been added in the discussion about the principle of the BumpHunter algorithm without going into the details. Please see the footnote at the bottom of page 3:
*“The method proposed by the BumpHunter algorithm relies purely on frequentist considerations, although there also exists different methods relying on a Bayesian approach [9].”*

However, we couldn’t find a relevant reference on the effect of going from a one dimensional bump hunt to a multidimensional one. Therefore, as you do not seem to insist too much on this, we would leave it as is.

2.
The choice of the negative sign for the significance of a deficit is indeed a convention.
In the case of a global p-value greater than 0.5, the sign of the significance changes and should become opposite to what is expected according to the convention.
We added the following sentence in the paragraph in order to clarify the behavior in the case of a search for a deficit (1st paragraph of page 4):
*“However, when considering a search for a deficit, the sign of the significance is flipped, so it will be negative in case the global p-value is smaller than 0.5 and positive otherwise.”*

3.
Our point in this paragraph is that the global p-value depends on the count of pseudo-data histograms that passes a given criteria (test statistic greater than a threshold).
If this count becomes very low, the behavior of the global p-value becomes inaccurate.
This problem is illustrated in another study that we have made recently and that is also available on arXiv: https://arxiv.org/abs/2211.07446
In the case when the count becomes exactly 0, the global significance goes to infinity.
This is a problem of available statistics, the higher the global significance, the more pseudo-data histograms are required.
We decided to rephrase the paragraph and add the paper mentioned above in the references.
Here is the part that has been modified (bottom of page 4):
*“This way of computing the global p-value is efficient to account for the look elsewhere effect, but has one limitation: the minimum non-null global p-value that can be computed depends on the number of pseudo-data distributions that were generated. When the count of pseudo-data distributions for which the test statistic passes the threshold ($N_{>data}$) becomes small, the obtained global p-value becomes inaccurate. In the extreme case when $N_{>data}$ becomes exactly 0, it causes the global significance to become infinite. [...]. Another approach to address this problem is discussed in [10].”*

4.
Using a mean+-68% interval (1 sigma) would indeed make more sense and this feature will be added to future releases of pyBumpHunter.
However we decided to keep this description in the paper since it corresponds to what the current stable release of pyBumpHunter does.

5.
Here we propose a new way to normalize the reference background histogram to the observed data in the case when the true normalization factor of the background is not known a priori (as for a floating normalization).
We made it more explicit in the text and also added a new figure (figure 8) in order to better illustrate the normalization procedure.
Here is the sentences we added to the text (page 10)
*“The figure 8 illustrates the principle of the side-band normalization procedure. Side-bands are used in the original proposal of the algorithm but do not treat the problem of the floating normalization of the reference background, as we propose.”*

Best regards,

Louis Vaslin, for the authors

---

## Round 3 · Referee Report · Mohamed Rameez (Referee 1) · 2023-3-28

Strengths

In addition to the strengths mentioned in the previous round of review, the inclusion of more comprehensive references makes the paper overall more readable now. The addition of figure 8 also improves the readability of the paper.

Weaknesses

The text itself is just an accompaniment to the code and tool.

Report

Perhaps this manuscript is better suited for Scipost physics codebases? I leave this to the editor to decide.

Requested changes

The author says "Using a mean+-68% interval (1 sigma) would indeed make more sense and this feature will be added to future releases of pyBumpHunter.
However we decided to keep this description in the paper since it corresponds to what the current stable release of pyBumpHunter does."

Perhaps this should be emphasized in text? That a nonstandard definition of uncertainties is being used.

  • validity: good
  • significance: good
  • originality: ok
  • clarity: high
  • formatting: good
  • grammar: excellent

Author:  Louis Vaslin  on 2023-04-06  [id 3560]

(in reply to Report 1 by Mohamed Rameez on 2023-03-28)
Category:
answer to question

Thank you again for your feedback.

We agree that one the main purpose of this paper is to present in details the features proposed in the pyBumpHunter package.
If it is more pertinent to have it in SciPost Physics codebases, we have no objections.

Concerning the nonstandard definition of uncertainties in section 2, we added a sentence in the caption of figure 4 :
"This is a nonstandard definition of uncertainties that will be modified in future release to cover a 68% confidence interval corresponding to 1σ."

Here is the link to the last version on arXiv : https://arxiv.org/abs/2208.14760v4

Best regards,

Louis Vaslin, for the authors

---

## Round 3 · List of Changes

Added the modification requested by the editorial report.

---

## Round 4 · Author Response

This version includes a minor modification following the recommendations of the reviewers.

You are currently on this page

Resubmission 2208.14760v4 on 6 April 2023

---

## Editorial Decision

published